# Overtemperature-protection intelligent molecular chiroptical photoswitches

Jiabin Yao [1], Wanhua Wu [1✉], Chao Xiao[1], Dan Su[1], Zhihui Zhong[1], Tadashi Mori [2] & Cheng Yang [1✉]

Stimuli-responsive intelligent molecular machines/devices are of current research interest due to their potential application in minimized devices. Constructing molecular machines/devices capable of accomplishing complex missions is challenging, demanding coalescence of various functions into one molecule. Here we report the construction of intelligent molecular chiroptical photoswitches based on azobenzene-fused bicyclic pillar[*n*]arene derivatives, which we defined as molecular universal joints (MUJs). The *Z/E* photoisomerization of the azobenzene moiety of MUJs induces rolling in/out conformational switching of the azobenzene-bearing side-ring and consequently leads to planar chirality switching of MUJs. Meanwhile, temperature variation was demonstrated to also cause conformational/chiroptical inversion due to the significant entropy change during the ring-flipping. As a result, photo-induced chiroptical switching could be prohibited when the temperature exceeded an upper limit, demonstrating an intelligent molecular photoswitch having over-temperature protection function, which is in stark contrast to the low-temperature-gating effect commonly encountered.

[1] Key Laboratory of Green Chemistry & Technology of Ministry of Education, College of Chemistry, State Key Laboratory of Biotherapy, West China Medical Center, and Healthy Food Evaluation Research Center, Sichuan University, Chengdu, China. [2] Department of Applied Chemistry, Osaka University, Suita, Japan. ✉email: wuwanhua@scu.edu.cn; yangchengyc@scu.edu.cn

Molecular machines/devices are miniaturized artificial devices, down to nano- to sub-nanoscales, which can accomplish large-amplitude mechanical motions by responding to input stimuli[1–3]. By virtue of an in-depth understanding of molecular motional physics, sophisticated design and organic synthesis, and stimulus-responsive molecular transformations, molecular machines/devices with complex structures and functions have been constructed[4], such as molecular shuttles[5–9], motors[10–13], pumps[14–16], and elevators[17]. These molecular machines/devices can switch among multi-isomeric states[18–20] by specifically responding to external stimuli, such as solvents[21,22], light[23–26], pH[27,28], redox[29–31], and chemical additives[32,33]. Temperature is a universal environmental factor. Unlike the above external stimuli, exerting a thermal effect on a molecular device without affecting its surrounding molecules is difficult. Lowering the temperature could lead to inhibition of some molecular physicochemical pathways, a phenomenon called low-temperature gating[34]. Increasing the temperature generally leads to improved reactivity and conversion pathways, and high-temperature gating has never been observed at the single-molecule level. Therefore, realizing overtemperature protection, which is vital for switching devices/power transistors, such as the stable operation of computers and devices and safety of power supplies, with a molecular device is obviously challenging. Herein, we report an overtemperature-protection function by integrating the thermo- and photoresponsive functions into one pillar[n]arene-based bicyclic pseudocatenanes, so-called molecular universal joints (MUJs)[30,35].

## Results

Azobenzene-bearing **MUJ1** and **MUJ2** (Fig. 1a) were synthesized by etherification of a triethylene glycol ether-modified trans-azobenzene derivative with dihydroxylated pillar[6]arene (P[6]) and pillar[5]arene (P[5]) derivatives, respectively, while **MUJ3** was obtained by etherification of the tetraethylene glycol ether-modified trans-azobenzene derivative with a dihydroxylated P[6] derivative[30]. The chemical structure of the MUJs was characterized by HR-mass spectrometry and NMR spectroscopic studies. As shown in Fig. 1b, the self-included "in" and self-excluded "out" conformers of the trans-MUJ enantiomers can interconvert quickly accompanied by the switching of chiral conformers that have clockwise-directed ($R_p$ conformer) and anticlockwise-directed ($S_p$ conformer) arrangement of the substituents, respectively[36,37]. However, for cis-MUJ, the large steric hindrance

of cis-azobenzene prevents the self-inclusion by the cavity of pillar[6]arene and inhibits the in–out interconversion.

The azobenzene moiety of the MUJs readily underwent cis/trans photoisomerization upon wavelength-selective photoirradiation. trans-**MUJ1** showed strong absorption peaks at 294 nm ($\varepsilon$ = 25,896 cm$^{-1}$ M$^{-1}$) and 365 nm ($\varepsilon$ = 17,093 cm$^{-1}$ M$^{-1}$), assignable to the transitions of the P[6] and trans-azobenzene moieties, respectively. Photoirradiation at 365 nm led to a rapid decrease in the absorption band at 365 nm (Fig. 2a) at which the ratio of the absorption spectrum of trans-isomer divided by that of cis-isomer reaches the maximum, accompanied by an increase in the longer wavelength peak at 450 nm due to the transformation to cis-azobenzene. The $^1$H-NMR spectral studies indicated that irradiation with a 365 nm LED lamp for 2 min led to a significant upfield shift of proton a of the azobenzene moiety (Fig. 2b). Meanwhile, proton b of the side ring-bearing hydroquinone unit showed a downfield shift while upfield shift could be observed with protons of several other hydroquinone units (Fig. 2b), suggesting a significant conformational change from self-included in to self-excluded out conformation upon the trans to cis isomerization, which released the shielding and deshielding effects unequivently exerted on pillar[6]arene subunits. trans-**MUJ1** was almost completely converted into cis-**MUJ1** in the photostationary state (PSS) under 365 nm photoirradiation (Fig. 3b). This was also confirmed by HPLC analyses, which gave a cis/trans ratio of 97.8/2.2 in the PSS under 365 nm irradiation (Supplementary Fig. 38). High cis/trans ratios (>93.5/6.5) were also observed with **MUJ2** and **MUJ3** (Supplementary Table 1) due to the overwhelmingly high absorption of the trans-MUJs at 365 nm. The lifetime of cis isomers of MUJs will be discussed later.

The spectrum of trans-**MUJ1** was almost entirely recovered under photoirradiation of cis-**MUJ1** at 510 nm (Fig. 2a, Supplementary Fig. 38). The trans/cis ratios were in the range of 4.4~12.0 in the PSS at 510 nm. However, the $^1$H-NMR spectra of cis-**MUJ1** ultimately returned to that of trans-**MUJ1** after being kept in the dark for one week (Fig. 2b), suggesting that trans-**MUJ1** is a thermodynamically stable species[38]. Detailed HPLC traces suggested that the complete conversion of cis-**MUJ1** to trans-**MUJ1** in the dark required more than 66 h at room temperature (Supplementary Fig. 41). In contrast, photoirradiation of cis-**MUJ1** at 510 nm led to the PSS within several minutes. $^1$H-NMR investigation of the photoisomerization between the trans- and cis-MUJs demonstrated that the photoisomerization is reversible and repeatable without destroying the chemical structure of the MUJs (Supplementary Figs. 80–82).

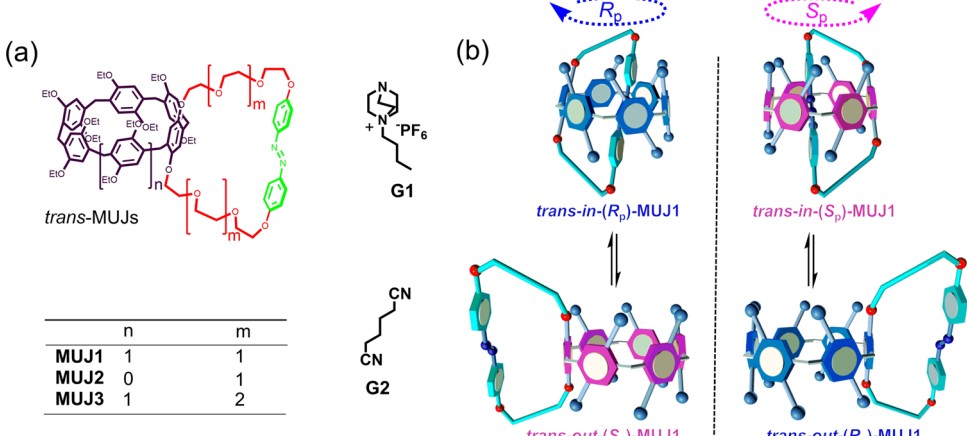

|  | n | m |
| --- | --- | --- |
| MUJ1 | 1 | 1 |
| MUJ2 | 0 | 1 |
| MUJ3 | 1 | 2 |

**Fig. 1 Chemical structures of MUJs and the *in–out* conformational switching of *trans*-MUJs. a** Chemical structures of **MUJ1**, **MUJ2**, **MUJ3**, **G1**, and **G2**. **b** Schematic diagram for the *in–out* equilibrium of the enantiomers of *trans*-**MUJ1**.

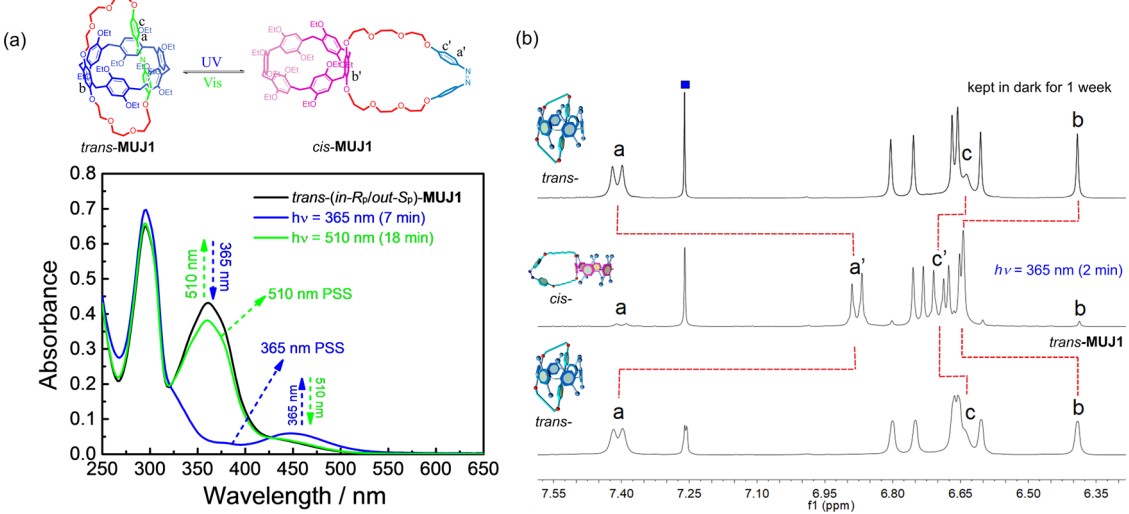

**Fig. 2 Photoisomerization of MUJ1 by wavelength-selective photoirradiation. a** UV-vis absorption spectra of *trans*-**MUJ1** (0.025 mM, chloroform) by alternating irradiation at 365 and 510 nm (xenon grating spectrometer). **b** ¹H NMR spectra of **MUJ1** (2 mM, 400 MHz, rt) in CDCl₃ before (bottom) and after (middle) irradiation at 365 nm for 2 min (LED illuminant) and then after (top) being kept in the dark for 1 week.

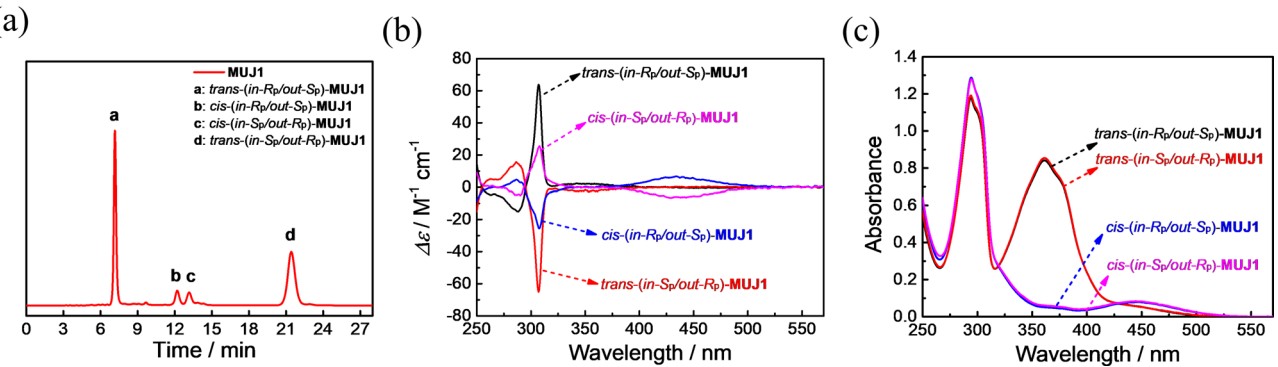

**Fig. 3 Chiral HPLC spectra and spectrum analysis of the enantiomers of MUJ1. a** Chiral HPLC trace of a mixture containing *trans*- and *cis*-**MUJ1** (tetrahydrofuran:*n*-hexane = 1:4, detected at 319 nm). **b** CD and (**c**) UV-vis spectra of enantiomeric pairs of *trans*-**MUJ1** and *cis*-**MUJ1** measured in the HPLC eluant (tetrahydrofuran:*n*-hexane = 1:4).

Each *trans*-MUJ is composed of a pair of isolatable planar chiral enantiomers (Fig. 1b)[37,39]. Successful chiral resolution of all three MUJs was obtained by chiral phase HPLC by using different mobile phases (Supplementary Figs. 35–37). As illustrated in Fig. 3a, four peaks, corresponding to two pairs of enantiomers of *trans*-**MUJ1** and *cis*-**MUJ1**, were well separated with certain eluent compositions, in which the first and fourth fractions were *trans*-**MUJ1**, and the second and third fractions were *cis*-**MUJ1**. Photoirradiation of the mixture at 365 nm led to an inversion of the relative intensities of the peaks a/d and b/c (Supplementary Fig. S42).

Circular dichroism (CD) spectra of isolated fractions were measured in the eluate (tetrahydrofuran:*n*-hexane = 1:4) at room temperature. The four fractions of **MUJ1** gave two sets of UV-vis spectra (Fig. 3c) and two pairs of mirror images in the CD spectra (Fig. 3b), in agreement with two pairs of enantiomers. The first- and third-eluted fractions of **MUJ1** showed positive CD extrema (CD_ex), and the second- and fourth-eluted fractions gave negative CD_ex (Fig. 3b). We have demonstrated that the negative CD_ex ca. 310 nm corresponds to the *Sp* (*Sp, Sp, Sp, Sp, Sp, Sp*) configuration of P[6] derivatives[30,35], and theoretical calculations confirmed that the *Sp* configuration of P[6] shows a negative CD_ex (Supplementary Fig. 157). However, since the azobenzene side ring could be either self-included (*in* conformer, Fig. 1b) or self-excluded (*out* conformer, Fig. 1b), the chirality and CD response of the

MUJs should critically depend on the conformers they adopt, and therefore, these CD spectral results did not immediately allow for assignment of the absolute configurations of these isomers.

**Absolute conformation of MUJs.** Competitive complexation experiments of the MUJs with **G1** or **G2** were performed to clarify the absolute configurations of the MUJs (Fig. 4a–c). In chloroform, the weak negative CD_ex of the **MUJ1** 1st fraction (Fig. 4a) was enhanced upon increasing the concentration of **G1**. This suggested that pushing out of the side ring by complexation of **G1** led to the *S_p* conformation of the pillar[6]arene core. The 1st fraction of **MUJ1** (Fig. 3a) was thus assigned to be *trans*-(*in-R_p/out-S_p*)-**MUJ1**, and the 4th fraction was *trans*-(*in-S_p/out-R_p*)-**MUJ1**. Photoisomerization of *trans*-(*in-R_p/out-S_p*)-**MUJ1** and *trans*-(*in-S_p/out-R_p*)-**MUJ1** led to the 2nd and 3rd fractions, respectively, indicating that the 2nd fraction is *cis*-(*in-R_p/out-S_p*)-**MUJ1** and that the 3rd fraction is *cis*-(*in-S_p/out-R_p*)-**MUJ1** (Supplementary Fig. 38). The absolute configuration of each fraction of **MUJ3** was assigned by the same procedure (Fig. 4c). **G1** is too bulky for the cavity of pillar[5]arene, and the absolute configuration of fractions of **MUJ2** was assigned based on the complexation experiment with **G2** (Fig. 4b). However, little CD spectral change could be observed with *trans*-(*in-R_p/out-S_p*)-**MUJ2**, even after adding a large excess amount of **G2**, suggesting

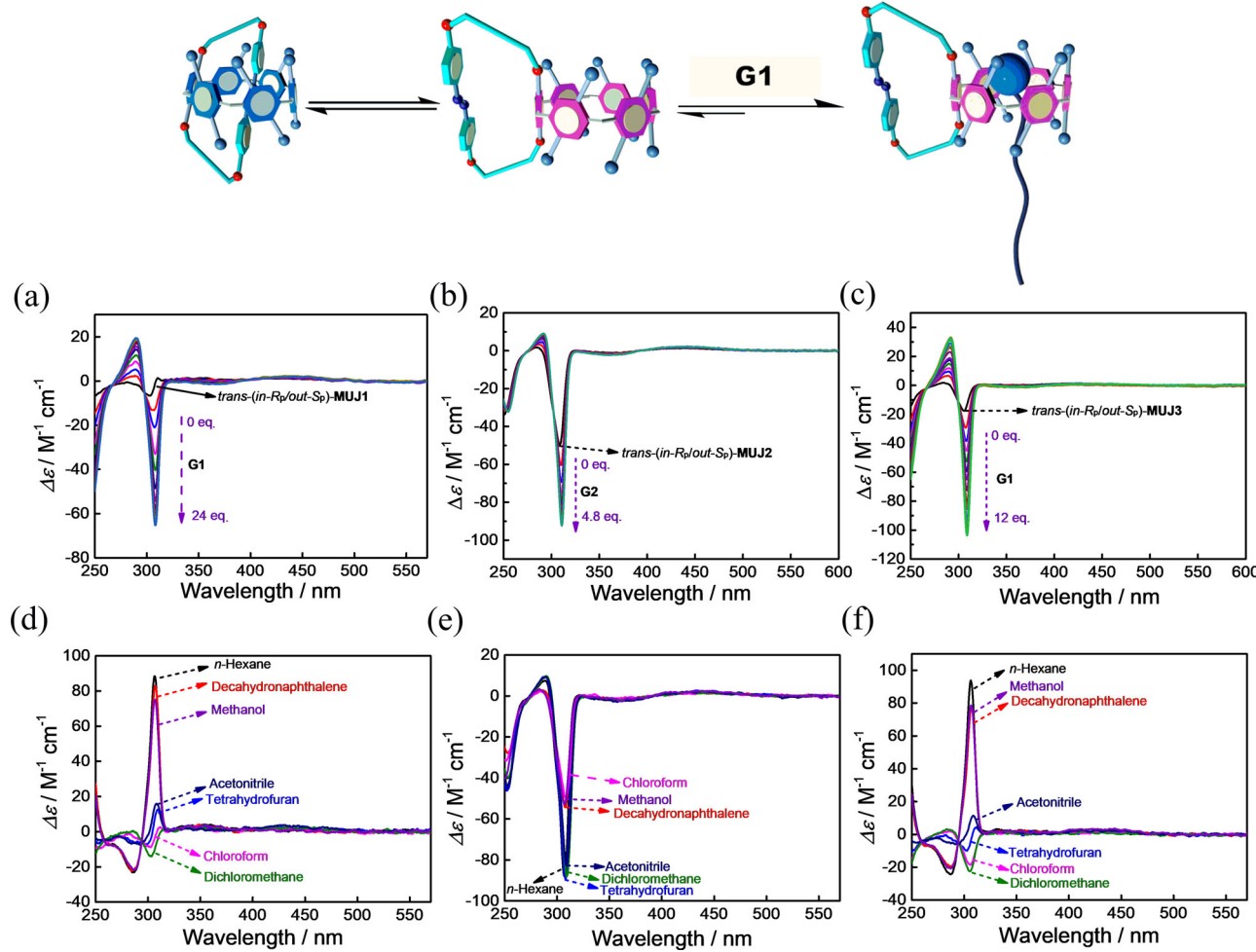

**Fig. 4 Solvent effects and competitive complexation of MUJs.** CD spectra of *trans-(in-R_p/out-S_p)*-**MUJ1** (**a**) upon increasing the concentration of **G1** in chloroform and (**d**) in different solvents; CD spectra of *trans-(in-R_p/out-S_p)*-**MUJ2** (**b**) with titrations of **G2** in chloroform and (**e**) in different solvents; CD spectra of *trans-(in-R_p/out-S_p)*-**MUJ3** (**c**) upon increasing the concentration of **G1** in chloroform at 25 °C and (**f**) in different solvents.

that *trans-(in-R_p/out-S_p)*-**MUJ2** almost completely adopts the *out* conformer (Supplementary Fig. 44). Based on the titration experiments (Supplementary Figs. 43, 45 and 47) the association constants $K_{assoc}$ of **G1** were determined as $(4.9 \pm 0.2) \times 10^3 \, M^{-1}$ for **MUJ1** and $(7.5 \pm 0.4) \times 10^3 \, M^{-1}$ for **MUJ3**, respectively, and **G2** showed a $K_{assoc}$ of $(8.0 \pm 0.8) \times 10^4 \, M^{-1}$ for **MUJ2**.

The CD spectra of the *trans-(in-R_p/out-S_p)*-MUJs were measured in various solvents (Fig. 4d–f), including methanol, acetonitrile, dichloromethane, chloroform, tetrahydrofuran, decahydronaphthalene, and *n*-hexane. As illustrated in Fig. 4d, the CD spectra of *trans-(in-R_p/out-S_p)*-**MUJ1** at room temperature were highly solvent-dependent: $CD_{ex}$ was strongly positive in the solvents *n*-hexane, decahydronaphthalene, and methanol and weakly positive in acetonitrile or tetrahydrofuran, indicating that a self-included (*in*) conformation dominated in all these solvents. However, $CD_{ex}$ was negative in chloroform or dichloromethane, suggesting that a self-excluded (*out*) conformer was adopted in these two solvents. The solvent-dependent CD spectra of *trans-(in-R_p/out-S_p)*-**MUJ3** were similar to those of *trans-(in-R_p/out-S_p)*-**MUJ1** (Fig. 4f). Such solvent effects reflect a delicate interaction balance between the *in* and *out* conformers, which involves binding of the solvent molecule(s), self-complexation of the side ring by the pillar[n]arene cavity, and solvation of the side ring. In particular, different solvation of the side ring and complexation of solvents by the pillar[n]arene cavity should be responsible for this distinct solvent dependence[35]. The strong positive $CD_{ex}$ observed in *n*-hexane, decahydronaphthalene and methanol should be ascribed to the unfavorable solvation effects of these solvents for the fused azobenzene moiety of **MUJ1** and **MUJ3**, which facilitated a shift of the *in–out* equilibrium to form self-complexation conformers. In contrast, the P[5]-based *trans-(in-R_p/out-S_p)*-**MUJ2** always presented strong negative $CD_{310}$ in all solvents examined, implying that the smaller cavity of P[5] hardly accommodated the azobenzene block of **MUJ2**, and the *out* conformation dominated in different solvents[40]. Furthermore, CD signals of *trans-(in-R_p/out-S_p)*-**MUJ2** in acetonitrile, dichloromethane, and *n*-hexane were stronger than in chloroform, decahydronaphthalene, and methanol, presumably due to the strong complexation of the formers with pillar[5]arene which fixes the orientation of each subunit.

**Light-driven chirality switching of MUJs.** *Trans-(in-R_p/out-S_p)*-**MUJ1** showed a strong positive $CD_{ex}$ in *n*-hexane at 20 °C due to the self-included *in-R_p* conformation. Photoirradiation of **MUJ1** at 365 nm led to a significant decrease in the absorption peak at 365 nm and an increase in the long-wavelength range absorption. Such absorption changes can commonly be used as readouts of azobenzene-based molecular switches. Interestingly, the strong positive $CD_{ex}$ of *trans-(in-R_p/out-S_p)*-**MUJ1** decreased and inverted to a strong negative CD signal upon irradiation at 365 nm, and the weak CD signal peak at 435 nm significantly increased. Moreover, the original CD spectrum could be recovered by photoirradiation

at 510 nm, at which the ratio of the absorption spectrum of *cis*-isomer divided by that of *trans*-isomer reaches the maximum (Fig. 6a and Supplementary Figs. 54–67). No fatigue was observed for the light-driven planar chirality switching between *trans-in-*($R_p$)-**MUJ1** and *cis-out-*($S_p$)-**MUJ1** after several tens of cycles (Fig. 6b).

Such a CD spectral change indicated a switching from *trans-in-*($R_p$)-**MUJ1** to *cis-out-*($S_p$)-**MUJ1**, demonstrating that the planar chirality of **MUJ1** can be switched by changing the irradiation wavelength. Such a photoinduced CD spectral change provides a powerful chiroptical switch that allows for distinctively determining on/off based on the positive/negative sign of $CD_{ex}$ rather than on the intensity, as commonly encountered in absorption and emission spectroscopic techniques. The rolling out of the side

ring upon the *trans* to *cis* isomerization seems reasonable because the *cis*-azobenzene block became too bulky to be accommodated in the cavity of P[6], i.e., the *trans*-azobenzene block preferred to be included in the cavity, while it was excluded upon conversion to the *cis*-configuration (Fig. 5). Alternating photoirradiation of *trans-*($in$-$R_p$/$out$-$S_p$)-**MUJ1** at 365 nm and 510 nm led to reversible switching of both the CD (anisotropy factor $g = \Delta\varepsilon/\varepsilon$, where $\varepsilon$ is the molar extinction coefficient at a particular wavelength, Fig. 6b) and UV-vis (Supplementary Figs. 56–67) spectra between their PSSs.

In contrast, *trans-*($in$-$R_p$/$out$-$S_p$)-**MUJ2** showed a strong negative $CD_{ex}$ in *n*-hexane at room temperature, suggesting that the *trans*-azobenzene moiety was located outside the P[5] cavity, which could be ascribed to the azobenzene moiety being too

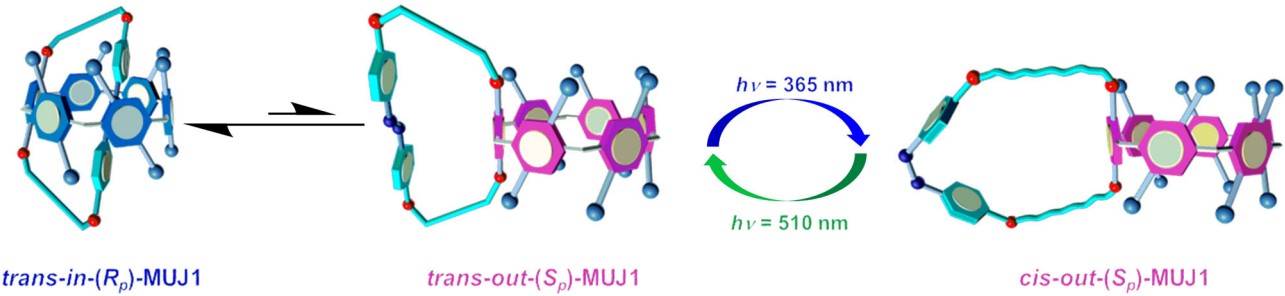

*trans-in-*($R_p$)-MUJ1  $\quad$  *trans-out-*($S_p$)-MUJ1  $\quad\quad$  *cis-out-*($S_p$)-MUJ1

**Fig. 5 Light-driven chirality switching of (*in-$R_p$/out-$S_p$*)-MUJ1.** Suppositional mechanism of light-driven chirality switching of (*in-$R_p$/out-$S_p$*)-**MUJ1**.

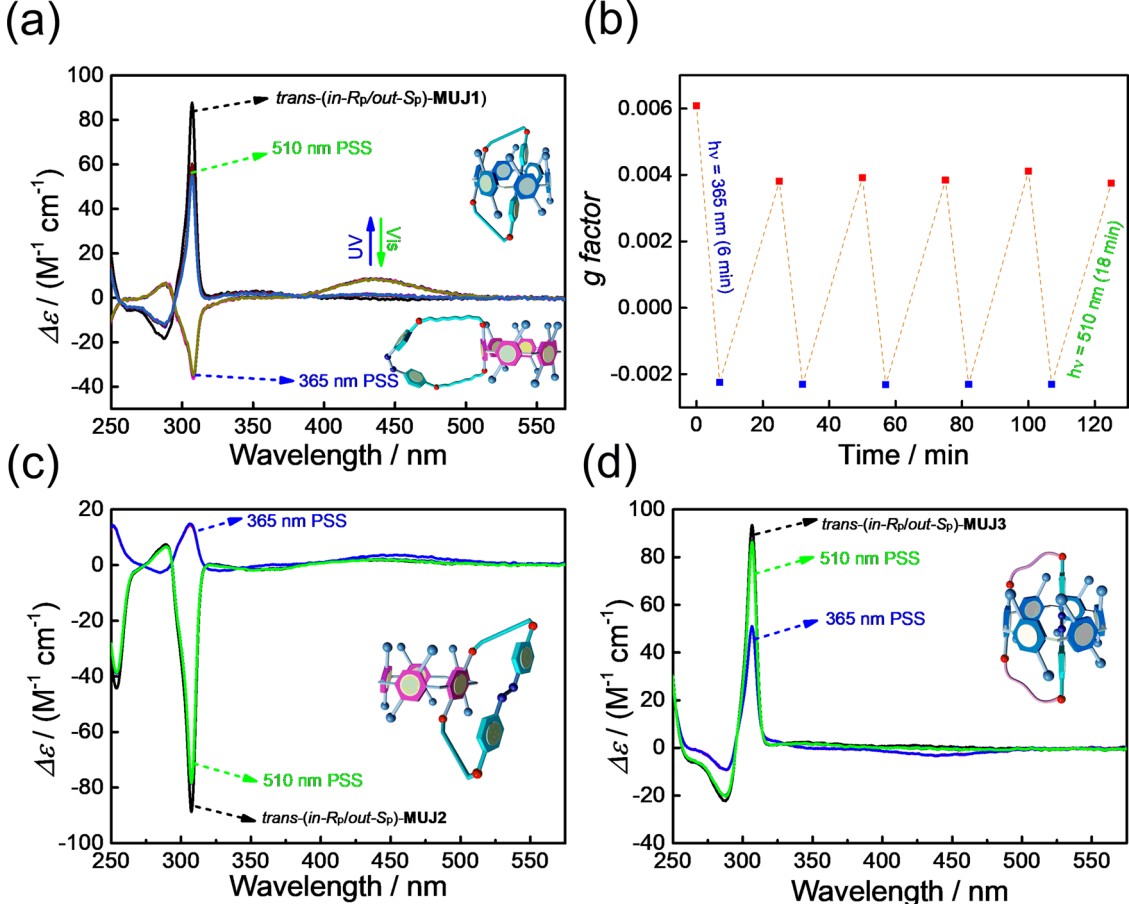

**Fig. 6 CD spectral change of the enantiomers of MUJs upon selective wavelength photoirradiation. a** CD spectra and (**b**) extremum changes (~310 nm) of the anisotropy factor (*g*) of *trans-*(*in-$R_p$/out-$S_p$*)-**MUJ1** upon alternating irradiation at 365 nm (blue signet) and 510 nm (red signet, xenon light source) in *n*-hexane. CD spectra of (**c**) *trans-*(*in-$R_p$/out-$S_p$*)-**MUJ2** and (**d**) *trans-*(*in-$R_p$/out-$S_p$*)-**MUJ3** upon alternating irradiation at 365 and 510 nm in *n*-hexane.

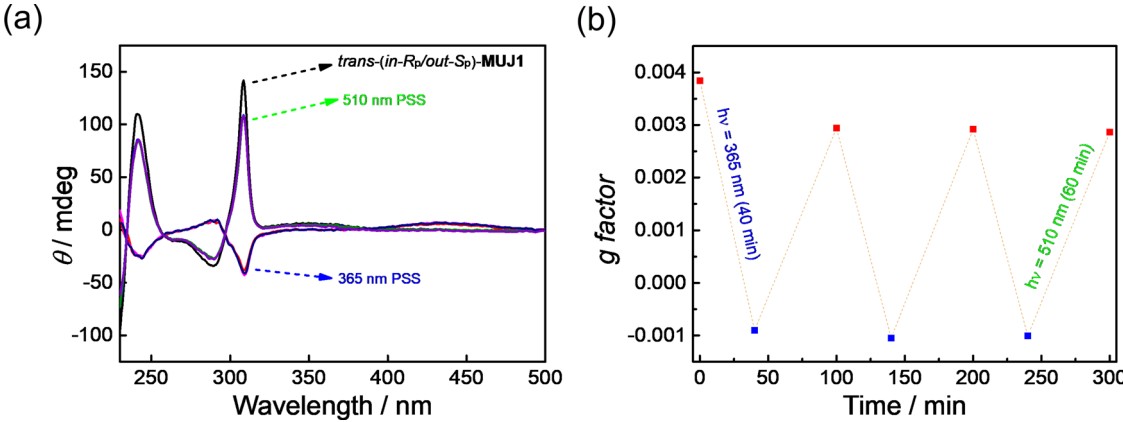

**Fig. 7 Light-driven chirality switching of (in-$R_p$/out-$S_p$)-MUJ1 in the coating film. a** CD spectra of the spin-coated film of *trans*-(*in*-$R_p$/*out*-$S_p$)-**MUJ1** under alternating irradiation at 365 and 510 nm (xenon light source). **b** Extremum changes (~310 nm) of the anisotropy factor g of *trans*-(*in*-$R_p$/*out*-$S_p$)-**MUJ1** upon alternating irradiation at 365 nm (blue signet) and 510 nm (red signet).

bulky to be comfortably accommodated by the P[5] cavity[41]. Photoirradiation of *trans*-(*in*-$R_p$/*out*-$S_p$)-**MUJ2** at 365 nm led to a significant decrease and almost inversion in CD$_{ex}$ (Fig. 6c and Supplementary Figs. 68–69), suggesting that the *out* reference became weaker. Molecular model studies of *trans*-**MUJ2** suggested that the side ring is stretched flat and is relatively rigid (Supplementary Fig. 84) due to the long strip structure of azobenzene, which prevents even partial inclusion of the side ring by the small pillar[5]arene cavity. However, the side ring in *cis*-**MUJ2** is much more flexible, which allows the triethylene glycol ether chain to be partially included in the P[5] cavity[35] (Supplementary Figs. 85–86). Indeed, $^1$H-NMR spectra of **MUJ2** recorded after 365 nm irradiation showed significantly upfield-shifted protons of the triethylene glycol ether chain, demonstrating self-inclusion of the fused glycol ether once *trans*-**MUJ2** transformed to *cis*-**MUJ2** (Supplementary Fig. 81). However, the *cis*-azobenzene block is too bulky for the P[5] cavity, which prevents complete chirality switching to the self-included conformer. The decreased negative CD$_{ex}$ caused by irradiation at 365 nm almost completely recovered to the initial state after irradiation at 510 nm (Fig. 6c and Supplementary Figs. 68–69).

*Trans*-(*in*-$R_p$/*out*-$S_p$)-**MUJ3**, which has a longer glycol ether chain and therefore a more flexible side ring than *trans*-(*in*-$R_p$/*out*-$S_p$)-**MUJ1**, also showed a strong positive CD$_{ex}$ (Fig. 6d) in *n*-hexane, suggesting that the *trans*-azobenzene block of **MUJ3** was self-included in the cavity. However, unlike the chiroptical inversion observed with **MUJ1**, the positive CD$_{ex}$ of **MUJ3** was significantly decreased under irradiation at 365 nm and recovered with irradiation at 510 nm. This phenomenon suggested that the side ring of **MUJ3** was only partially excluded from the pillar[6]arene cavity when transformed to *cis*-azobenzene (Supplementary Figs. 72–75). We ascribed this to the more flexible side ring of **MUJ3**, which allows the glycol ether chain to be partially self-accommodated in the P[6] cavity when the *trans*-azobenzene isomerizes to the *cis*-azobenzene, thus hampering the chirality switching of *trans*-(*in*-$R_p$/*out*-$S_p$)-**MUJ3**. The above result indicated that the steric effect was critically affected by the cavity size and the chain length in the present system, and appropriate size matching between the cavity and the side ring as well as the rigidity of the side rings is pivotal to the light-driven chirality switching behavior.

Interestingly, the light-driven chirality switching of *trans*-(*in*-$R_p$/*out*-$S_p$)-**MUJ1** could also be observed in the coating film. A film of *trans*-(*in*-$R_p$/*out*-$S_p$)-**MUJ1** was prepared by spin-coating and then kept under vacuum to dry for 24 h. The film showed the

strong positive CD$_{ex}$ of *trans*-(*in*-$R_p$/*out*-$S_p$)-**MUJ1**, which changed to a strong negative CD$_{ex}$ when irradiated at 365 nm but recovered to the positive CD$_{ex}$ under irradiation at 510 nm (Fig. 7a). The reversible light-driven chirality switching of *trans*-(*in*-$R_p$/*out*-$S_p$)-**MUJ1** could be repeated many times (Fig. 7b), indicating that the *trans*/*cis* isomerization of the azobenzene block can effectively trigger mechanical rolling in/out of the side ring[42]. Such light-driven planar chirality switching of the unimolecular system in the coating film makes **MUJ1** a promising candidate for photoresponsive switch materials.

**Overtemperature protection for light-driven chirality switching**. Self-sensing and a feedback ability for temperature perturbances are critically important for intelligent microdevices, having potential applications in, for example, microenvironment temperature sensing and overtemperature protection of microcircuits. While inhibition of certain molecular chemical and physical processes by reducing the temperature is an often-observed phenomenon, inhibiting on/off molecular switching by increasing the temperature is challenging, particularly at the single-molecule level. Variable temperature CD (VT CD) spectra of *trans*-(*in*-$R_p$/*out*-$S_p$)-MUJs were examined in various solvents (Supplementary Figs. 87–107). Increasing the temperature led to a decrease in the CD$_{ex}$ of *trans*-(*in*-$R_p$/*out*-$S_p$)-**MUJ2** in all solvents examined (Fig. 8b and Supplementary Figs. 94–100), indicating that the *out* conformer *trans-out*-($S_p$)-**MUJ2** was the main conformer in the temperature range examined. However, the CD spectra of *trans*-(*in*-$R_p$/*out*-$S_p$)-**MUJ1** and *trans*-(*in*-$R_p$/*out*-$S_p$)-**MUJ3** exhibited significant temperature dependence. For *trans*-(*in*-$R_p$/*out*-$S_p$)-**MUJ1**, as an example, the negative CD$_{ex}$ in tetrahydrofuran at 50 °C was reduced upon decreasing the temperature and finally inverted to give a positive CD$_{ex}$ at 5 °C (Fig. 8a), demonstrating a conformational transformation from *trans-out*-($S_p$)-**MUJ1** to *trans-in*-($R_p$)-**MUJ1**. The UV-vis spectra of the *trans*-MUJs were hardly changed by the temperature variation (Supplementary Figs. 87–107), indicating no *trans* to *cis* isomerization of the azobenzene blocks with the temperature variation. Raising the temperature back to 50 °C led to a complete recovery of the initial CD spectrum of *trans*-(*in*-$R_p$/*out*-$S_p$)-**MUJ1**. The *in* conformer is favored at low temperatures, which seems reasonable because the *in* conformer should be stabilized by van der Waals contact, C–H···O, π−π, and dipole–dipole interactions between the P[6] cavity and the side ring but is entropy disfavored due to the loss of the motional and rotational

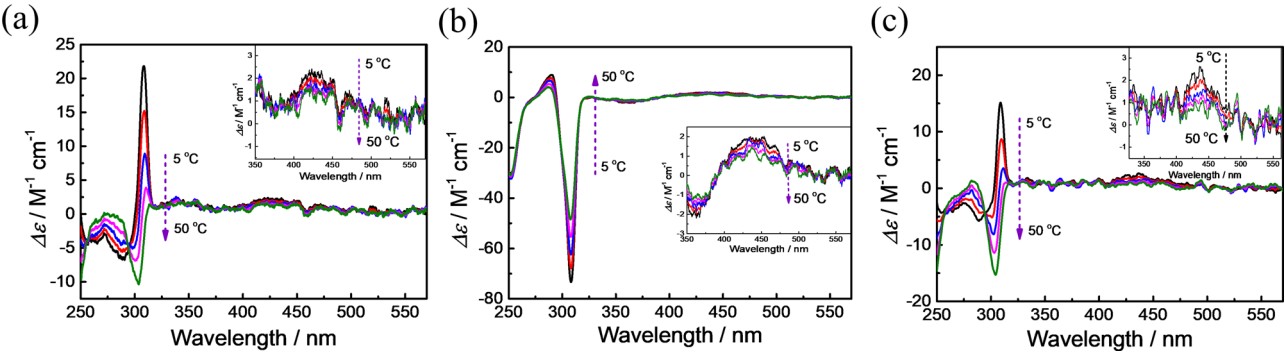

**Fig. 8 Variable temperature CD spectra of the enantiomers of MUJs.** VT CD spectra of (**a**) *trans*-(*in*-$R_p$/*out*-$S_p$)-**MUJ1**, (**b**) *trans*-(*in*-$R_p$/*out*-$S_p$)-**MUJ2**, and (**c**) *trans*-(*in*-$R_p$/*out*-$S_p$)-**MUJ3** in tetrahydrofuran at 5 °C (black line), 15 °C (red line), 25 °C (blue line) 35 °C (magenta line) and 50 °C (green line), respectively.

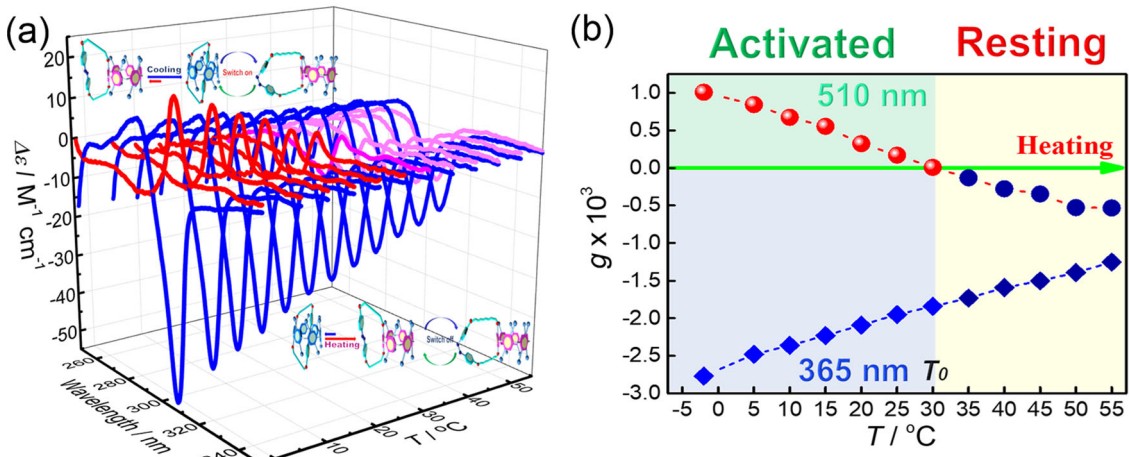

**Fig. 9 Temperature regulation of the light-driven CD spectral inversion of enantiomeric MUJ1. a** VT CD spectra of (*in*-$R_p$/*out*-$S_p$)-**MUJ1** in the PSS at 365 nm (blue line) and 510 nm (red, magenta, and pink lines) in tetrahydrofuran. The CD spectra of (*in*-$R_p$/*out*-$S_p$)-**MUJ1** with positive CD extrema ($CD_{ex}$) at ca. 310 nm is shown as the red line, while that with negative $CD_{ex}$ is shown as the pink line and that between the two is shown as the magenta line. **b** VT anisotropy factor changes at 308 nm of (*in*-$R_p$/*out*-$S_p$)-**MUJ1** in the PSS at 365 nm (diamond) and 510 nm (circle) in tetrahydrofuran. The blue signet presenting a negative *g* factor and the red signet presenting a positive *g* factor.

freedom of both the side ring and P[6] units. Temperature-driven chirality inversions were observed in tetrahydrofuran, chloroform, dichloromethane, and acetonitrile for **MUJ1** (Fig. 8a and Supplementary Figs. 90–92) and in tetrahydrofuran and acetonitrile for **MUJ3** (Fig. 8c and Supplementary Fig. 106). Such temperature-driven chiroptical switching demonstrated a delicate dynamic balance established by the significant entropy differences between the *in* and *out* conformers. It, therefore, provided a powerful tool to regulate the on/off switching of the MUJs and to implement sensing and feedback of microenvironmental temperature perturbances. Based on the VT CD spectral changes[43], the entropy ($\Delta\Delta S$) and enthalpy ($\Delta\Delta H$) changes for the *in*/*out* conformational switching of **MUJ1** in $CHCl_3$ were estimated as $-16.5$ J mol$^{-1}$ K$^{-1}$ and $-4.5$ KJ mol$^{-1}$, respectively. This demonstrates that the *out*-to-*in* conformational switching is enthalpically favorable but disfavored entropically.

The unique temperature and light dual-responsive property of **MUJ1** endow it with the potential to serve as a temperature-responsive molecular photoswitch. Irradiating *trans*-(*in*-$R_p$/*out*-$S_p$)-**MUJ1** at $-2$ °C with light at 365 and 510 nm caused definite $CD_{ex}$ sign inversion (Fig. 9a), demonstrating the operation of light-driven switching. However, at 55 °C, *trans*-(*in*-$R_p$/*out*-$S_p$)-**MUJ1** consistently showed a negative $CD_{ex}$ upon irradiation at both 365 and 510 nm (Fig. 9a), indicating silencing of the chirality

inversion at high temperatures. Detailed investigation of the temperature-dependent photoswitching behavior (Fig. 9a) revealed a critical temperature $T_0$ of 30.3 °C (Fig. 9b), below which $CD_{ex}$ exhibited positive/negative switching upon alternating photoirradiation with light at 365 nm and 510 nm and above which $CD_{ex}$ consistently exhibited a negative sign regardless of the irradiation wavelength employed. Therefore, the VT CD spectra of (*in*-$R_p$/*out*-$S_p$)-**MUJ1** in the PSS at 365 nm were investigated (Supplementary Figs. 113–118) to gain insight into the temperature-regulated light-driven chirality switching. As shown in Fig. 9b, (*in*-$R_p$/*out*-$S_p$)-**MUJ1** in the PSS at 365 nm presented a consistent negative *g* factor in all of the examined temperature regime, while (*in*-$R_p$/*out*-$S_p$)-**MUJ1** in the PSS at 510 nm presented a positive *g* factor in the low-temperature regime ($T < 30.3$ °C) and a negative *g* factor in the high-temperature regime ($T > 30.3$ °C). This phenomenon indicated that temperature can serve as an effective external stimulus to regulate the light-driven chirality switching of (*in*-$R_p$/*out*-$S_p$)-**MUJ1**, which can be switched on by cooling and off by heating. As discussed above, the regulatory mechanism should be achieved by the synergistic effects between the two independent mechanisms, i.e., the temperature-regulated *in*–*out* equilibrium and the light-controlled photoisomerization of the azobenzene block (Supplementary Fig. 156).

**Table 1 Critical temperature ($T_0$) of (in-$R_p$/out-$S_p$)-MUJ1 in different solvents.**

| Solvent | $T_0$/°C (308 nm) |
|---|---|
| Tetrahydrofuran | 30.3 |
| Acetonitrile | 47.6 |
| Chloroform | 29.7 |
| Acetonitrile:tetrahydrofuran = 1:1 | 8.8 |
| Acetonitrile:tetrahydrofuran = 1:19 | 31.0 |
| Acetonitrile:tetrahydrofuran = 19:1 | 34.0 |

**Adjustable critical temperature of switching**. Furthermore, the critical temperature ($T_0$) for the regulation of the light-driven chirality switching can be adjusted by changing the solvents or solvent composition (Table 1). For example, the critical temperature $T_0$ was improved to 47.6 °C in acetonitrile but decreased to 29.7 °C in chloroform. Moreover, by using a mixture of acetonitrile and tetrahydrofuran as a solvent, $T_0$ could be conveniently adjusted from 8.8 to 34 °C simply by changing the solvent composition. This unambiguously expands the application domains and improves the regulatory capabilities of the intelligent chirality switch.

## Discussion

In summary, we demonstrated an intelligent molecular switch that shows a temperature- and photoresponsive on/off switch function. Such orthogonal control allows photoinduced rolling in/out and chiroptical switching to be prohibited when the temperature exceeds an upper limit, realizing overtemperature-protection intelligent molecular photoswitches. The upper-temperature limit could be conveniently adjusted by manipulating the solvent composition. This study realized the challenging high-temperature-gating effect at the molecular level and represents a prominent step forward for constructing an intelligent molecular machine/device capable of performing complex functions.

## Methods

A Chiralpak IA column was used for chiral separation. The 365 and 510 nm light sources for photoresponsive CD spectral measurement were used with a grating spectrometer on a JASCO FP 8500 fluorescence spectrometer. A 365 nm LED spot curing system was used for photoresponsive $^1$H-NMR measurements. CD spectra were recorded as $\theta$ in millidegrees. The anisotropy factor $g$ was calculated using the equation $g = \Delta\varepsilon/\varepsilon$, where $\varepsilon$ is the molar extinction coefficient at a particular wavelength. The molar ellipticity was obtained by the formula $\Delta\varepsilon = \theta/32980cl$, where $l$ is in cm.

## Data availability

The data are available upon reasonable request.

## Code availability

The codes are available upon reasonable request.

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

## Acknowledgements

We acknowledge the support of this work by the National Natural Science Foundation of China (Nos. 92056116, 21871194, 21971169), National Key Research and Development Program of China (No. 2017YFA0505903), Key R & D project of Science & Technology Department of Sichuan Province (2019YJ0160, 2019YJ0090, and 2017SZ0021), CREST, JST, Japan (Grant No. JPMJCR2001), China Postdoctoral Science Foundation (No. 2019M653393), Comprehensive Training Platform of Specialized Laboratory, College of Chemistry and Prof. Peng Wu of Analytical & Testing Center, Sichuan University.

## Author contributions

C.Y. initiated the project. J.Y., W.W., and C.X. conceived and designed the experiments, analyzed the data, and prepared the manuscript, with input from all the authors. J.Y. conducted the chemical synthesis. D.S and Z.Z. gave advice on CD and UV-Vis spectral analysis. T.M. conducted theoretical calculations of the CD spectra.

## Competing interests

The authors declare no competing interests.
