## [Peer Review File · Nature Communications]

REVIEWER COMMENTS

Reviewer #1 (Remarks to the Author):

In this nice manuscript, the authors report an intelligent molecular chiroptical photoswitch by integrating a pillar[6]arene and an azobenzene side-ring, which showed light-driven reversible chirality switching both in the solid-state and in solution, even to exhibit overtemperature-protection function in solution. The chirality switching behavior of the photoswitch can be orthogonally controlled by wavelength-selective photoirradiation and temperature regimes. The photoisomerization and conformational changes are fully reversible, which is demonstrated on the basis of CD spectroscopy, NMR spectra, and HPLC analysis. The authors also found that a high-temperature gating effect of such intelligent molecular chiroptical photoswitch, with the critical temperature being conveniently adjustable by manipulating the solvent composition. Overall, I am of the opinion that the light-driven conformational switching (and associated g-factors) with overtemperature-protection function is ingenious and of sufficient novelty to warrant publication in Nature Communications. The manuscript describes interesting findings and the overtemperature-protection effects which might hold promise for design future dynamic and intelligent molecular machines as well as (supra-)molecular systems. This manuscript can be accepted for publication after considering the following minor suggestions and questions:

- 1) The authors should clarify why 365 nm and 510 nm light sources were chosen to achieve the conformational switching? (the absorbance of the MUJs at 510 nm is very weak)
- 2) The component ratio of cis- or trans-MUJs in the PSSs calculated based on the HPLC analysis should be provided.
- 3) The reason why the mixed solvent (THF : n-H = 1:4) rather than a pure solvent component was used for spectral measurement should be clarified. Eg. the UV-vis spectra in Figure 1a and Figure 2c, and the CD spectra in Figure 2b.
- 4) In the competitive complexation experiments (Figure 3b), trans-(in-Rp/out-Sp)-MUJ2 almost completely adopts the out conformer in solution. Although the P[5]-based trans-(in-Rp/out-Sp)-MUJ2 always presented strong negative CD₃₁₀ in different solvents (Figure 3e), the relatively large intensity differences of the CD signals are shown in CHL, DECA and MeOH comparing to other solvents (AN, DCM, THF, n-H). If possible, please give a reasonable explanation for this phenomenon in the main text.
- 5) In the legend of Figure 7, "(c) Schematic diagram for the temperature-limit-protected light-driven chirality switch (in-Rp/out-Sp)-MUJ1" should be removed.

Reviewer #2 (Remarks to the Author):

This is an interesting manuscript that describes a molecular switch based on pillar[n]arenes with an appended diazo-benzene containing loop. This structure allows photoinduced chiroptical switching (known property of this design) to be prohibited when the temperature exceeds a critical limit. The authors describe this as the first realization of overtemperature protection for a molecular photoswitch. The upper-temperature limit was shown to be adjustable by manipulating the solvent. The experiments are detailed and complete, while the rationales/conclusions are appropriate. I recommend publication after attention to the following comments.

1. The introductory paragraph describes examples and defines molecular machines in rather broad terms. Some of this is not very precise. For example, machines do work while switches (like those in

- this article) do not; this is a very important distinction. This paragraph and the associated references should be checked and reworded so as not to perpetuate any incorrect or vague terminology.
2. I encourage updating of references for certain topics, e.g. Molecular pumps J. Am. Chem. Soc. 2020, 142, 34, 14443 and J. Am. Chem. Soc. 2019, 141, 44, 17472, Molecular shuttles Nat. Chem. 2015, 7, 514 and Nat. Chem. 2018, 10, 625.
 3. When varying temperature for measurements, this should be referred to as variable temperature and abbreviated as VT (e.g. VT NMR).
 4. Fig. 7, there is no part (c) as described in the caption.
 5. HRMS is high resolution mass spectrometry – not spectroscopy.
 6. Scheme-1: In (a) the chemical structure would be easier to view without the background graphic (which is not described or particularly relevant), please remove. This will make it easier also to understand the relationship between the chemical structure and the cartoons used in (b) and the rest of the article.
 7. Figure 1: In (a) the m and brackets should be removed from the cis structure. The trans-cis conversion is shown as out-out in (a) but for (b) the cartoons show trans as in? Which is correct? Is the in-out a fast (NMR timescale) equilibrium? This equilibrium (Scheme-1b) should be described in more detail first before the cis trans isomerization. For example, the ¹H NMR spectra shown in (b) are certainly indicative of a reversible cis-trans isomerization but cannot (by themselves) distinguish between trans in or out. I understand that the HPLC resolution and CD spectral assignments address the 4 possible isomers, but it is not clear how these mixtures manifest themselves in the simple ¹H NMR shown in (b).
 8. It would be appropriate to have the host guest interactions for G1 and G2 be more quantitative. What are the association constants for these interactions? Can the K_{assoc} for the in versus out isomers be measure? How do the K_{assoc} values in different solvents correlate to the easy of switching.
 9. Do not abbreviate n-hexane to n-H, or acetonitrile to AN, likewise CHL, DCM etc. Use chemical formulae or accepted abbreviations e.g. MeCN for acetonitrile.
 10. Describing switching in a thin film as the switching in the solid-state is not accurate. This is an ill-defined, amorphous film not a well packed solid.
 11. Is Fig.7 not misleading? The chemical drawings only show cis as out and trans as in, but these are equilibria and the T dependence is actually the shifting of this equilibrium with change in T. Please define coloured traces.
 12. Perhaps g factor could be defined for those not familiar with this treatment of CD spectra.

Reviewer #3 (Remarks to the Author):

In this publication by Yang et al., an inherently chiral pillar[n]arene macrocycle (R/S) is strapped by an azobenzene photoswitch (cis/trans), a unit which can either bind inside or outside (in/out) the macrocycle's cavity. As indicated by the brackets, this system contains multiple switching components and their relationship to one another are explored in detail through this work. The key findings of this work are:

1. The isomer of the azobenzene (cis/trans) influences azobenzene binding inside the cavity
2. The inclusion of the azobenzene strap (as trans) can be controlled by light and switches the chirality of the pillar[n]arene (from S to R)
3. MUJ1 is the 'goldlocks' system, with optimal sterics and flexibility for effective switching
4. Temperature is also a stimulus because the light-induced chiral switch only occurs within a strict temperature range, i.e. the system is 'on' at low T and 'off' at high T

5. This 'temperature gating' (i.e. turning the system off at high T) is novel and advantageous because typically a higher T will enhance dynamic processes (turn on). The authors ascribe this to 'over temperature' protection.

In general, I agree with the authors on the novelty of the above. However, there are some key citations (one missing) that overlap and are relevant to the key findings above. Missing citations must be included and all should be better contextualised for this work. I have numbered these in relation to the above:

1. Differences in cis/trans azobenzene binding by pillar[n]arenes has been documented in this publication: *J. Am. Chem. Soc.* 2012, 134, 20, 8711–8717. Note that this is for a host-guest system and not a '[1]catenene' system described here.
2. Chirality switching of pillar[n]arene has been achieved through guest binding (*Chem. Commun.*, 2020,56, 8424-8427 *missing*), solvent binding (*Angew. Chem. Int. Ed.* 2013, 52, 8111 –8115) and acid/base (*J. Am. Chem. Soc.* 2020, 142, 46, 19772–19778). To the best of this reviewers knowledge the work here is the first example of light switchable chirality in a pillar[n]arene.

I am confident that both supramolecular and chiroptic communities will find this work interesting. The ability to switch chirality using a convenient achiral stimuli (light/temperature) is clearly important. Furthermore, as the authors note, the chirality switching results in a switch in CD sign which is more advantageous than intensity changes. This widens the applicability of this work to, for example, sensing and organic materials communities.

Overall, the work performed is convincing and the authors should be congratulated on the thoroughness of this study. However, this reviewer would ask that the following experiments be considered in order to strengthen the conclusions.

1. In lines 118-119 authors present the logical conclusion that trans-azobenzene binds in the cavity but cis cannot. Can the authors confirm this through host-guest binding studies, using an azobenzene as a discrete guest molecule for the cavities of the pillar[n]arenes? This work would link to the previous publication in this area: *J. Am. Chem. Soc.* 2012, 134, 20, 8711–8717. In connection with line 183 and following on from the above, the authors should perform variable temperature analysis of the binding of cis/trans-azobenzene guests to the pillar[n]arene hosts to elucidate the thermodynamic contributions to binding. This would provide clear evidence to support their conclusion that an entropic penalty is imposed upon guest binding. This is an important experiment because this entropic barrier is key to the temperature gating of the chiroptical switching behaviour
2. The authors conclude that the conformation of the system is solvent dependent (lines 89-90). Therefore it is important that the UV-Vis and NMR data reported in Fig 1 are also performed in the same solvent for accurate comparisons, e.g. UV-Vis to be performed in CHCl₃. Similarly, line 93 discusses the potential for 'binding of the solvent molecules'. Do the authors have evidence for DCM/CHCl₃ encapsulation within the pillar[n]arene instead of the thread? Perhaps the above UV-Vis and NMR spectroscopic experiments suggested above could shed light. Furthermore, reference to the following is essential: *Angew. Chem. Int. Ed.* 2013, 52, 8111 –8115 and *Chem. Commun.*, 2020,56, 8424-8427.
3. Is it possible to conduct the HPLC shown in Fig 2 a with the cis-isomerised azobenzene to show an inversion of the peak intensities i.e. b,c >>> a,d.

Some further points that are more minor but would greatly improve the publications clarity for the reader are listed below:

- Line 9: I think 'large-amplitude' should be added when deifying the molecular motion in molecular machines
- Scheme 1: Please indicate the planar chirality that arises from a pillar[n]arene on the cartoon (e.g. an arrow?). It might not be instantly obvious where the source of chirality is in the system and to separate it from the cis/trans geometry of the azobenzene.
- Lines 36-38: The ¹H NMR shifts are indicating more than just cis-trans isomerization. It is important to clarify that it indicated major changes in the molecule's conformation i.e. in/out
- Line 40: Clarify to the reader that the lifetime of cis isomer is discussed later
- Figure 1: There should be consistency in representation of the major conformation of the macrocycle in cis/trans isomer i.e. in trans the azobenzene is included in the macrocycle and so should be indicated in Fig 1a structure (as it is in Fig 1b)
- Line 53: The planar chirality arising from the pillar[n]arene is highlighted but should be indicated to the reader earlier in the manuscript (e.g. lines 26-28)
- Line 71: It would be helpful for the reader to see a cartoon of how guest binding leads to the strap being ejected i.e. favouring an out conformation
- Line 90: For clarity please adjust to read 'conformation dominated in all these solvents'
- Fig 4 would benefit from a cartoon to show the reader what conformational and chiral changes these CD spectra are evidencing
- Line 119: Supplementary Scheme 2 should be brought into the manuscript at this point to add clarity for the reader.
- Line 180 (and all solvent abbreviations): Please can solvents be written in full e.g. CHL = CHCl₃ or chloroform

The experimental details provided in the supporting information are detailed and thorough, suggesting that this work could be reproduced.

There is considerable interest in the development of chiroptical switches that can be controlled using facile external stimuli such as light. This publication may influence thinking in this area by asking researchers to consider the influence of entropy and thus temperature as a stimulus that is competitive or, perhaps in the future, can be harnessed.

RESPONSE TO REVIEWER COMMENTS

Reviewer #1 (Remarks to the Author):

In this nice manuscript, the authors report an intelligent molecular chiroptical photoswitch by integrating a pillar[6]arene and an azobenzene side-ring, which showed light-driven reversible chirality switching both in the solid-state and in solution, even to exhibit overtemperature-protection function in solution. The chirality switching behavior of the photoswitch can be orthogonally controlled by wavelength-selective photoirradiation and temperature regimes. The photoisomerization and conformational changes are fully reversible, which is demonstrated on the basis of CD spectroscopy, NMR spectra, and HPLC analysis. The authors also found that a high-temperature gating effect of such intelligent molecular chiroptical photoswitch, with the critical temperature being conveniently adjustable by manipulating the solvent composition. Overall, I am of the opinion that the light-driven conformational switching (and associated g-factors) with overtemperature-protection function is ingenious and of sufficient novelty to warrant publication in Nature Communications. The manuscript describes interesting findings and the overtemperature-protection effects which might hold promise for design future dynamic and intelligent molecular machines as well as (supra-)molecular systems. This manuscript can be accepted for publication after considering the following minor suggestions and questions:

We thank this reviewer for his/her highly positive comments.

1) The authors should clarify why 365 nm and 510 nm light sources were chosen to achieve the conformational switching? (the absorbance of the MUJs at 510 nm is very weak)

Action:

The light at the wavelengths of 365 nm and 510 nm were chosen for the photoisomerization of azobenzene, which was based on the absorption spectra of *trans*- and *cis*- MUJs to thus render maxima of the ratios of *trans/cis* and *cis/trans*. By applying the light of these wavelengths, we could obtain high *trans*-to-*cis* and *cis*-to-*trans* photoisomerization conversion.

Page 3 lines 38-39: “at which the ratio of the absorption spectrum of *cis*-isomer divided by that of *trans*-isomer reaches the maximum”, and page 8 lines 128-129: “at which the ratio of the absorption spectrum of *trans*-isomer divided by that of *cis*-isomer reaches the maximum” have been added.

2) The component ratio of *cis*- or *trans*-MUJs in the PSSs calculated based on the HPLC analysis should be provided.

Action:

The component ratios of *cis*- or *trans*-MUJs in the PSSs were calculated based on the HPLC analysis and were listed in the supplementary table 1: *cis/trans* = 97.8/2.2, 93.6/6.4, 94.2/5.8 were obtained under 365 nm irradiation for **MUJ1**, **MUJ2**, and **MUJ3**, respectively, and *trans/cis* = 81.5/18.5, 91.5/8.5, 92.3/7.7 were obtained under 510 nm irradiation.

3) The reason why the mixed solvent (THF : *n*-H = 1:4) rather than a pure solvent component was

used for spectral measurement should be clarified. Eg. the UV-vis spectra in Figure 1a and Figure 2c, and the CD spectra in Figure 2b.

Since the interconversion between *cis*-MUJs and *trans*-MUJs can occur even at room temperature, the spectral measurements of *trans* and *cis*-**MUJ1** were carried out immediately after HPLC separation to obtain spectra of pure *cis*-MUJs and *trans*-MUJs. Therefore, the HPLC fraction that using the eluent (THF : *n*-H = 1:4) was used directly for the spectral measurements.

Action: The UV-Vis spectra in Fig 1 have been replaced by the UV-Vis spectra that performed in chloroform to keep consistent with the ¹H NMR spectra in Fig 1b. The relevant discussion about the photoisomerization and chiral properties of MUJs based on the spectra (Fig 2b, 2c) tested in the mixed solvent is consistent with the conclusion obtained with the pure solvent.

4) In the competitive complexation experiments (Figure 3b), *trans*-(*in*-R_p/*out*-S_p)-**MUJ2** almost completely adopts the out conformer in solution. Although the P[5]-based *trans*-(*in*-R_p/*out*-S_p)-**MUJ2** always presented strong negative CD₃₁₀ in different solvents (Figure 3e), the relatively large intensity differences of the CD signals are shown in CHL, DECA and MeOH comparing to other solvents (AN, DCM, THF, *n*-H). If possible, please give a reasonable explanation for this phenomenon in the main text.

Action:

The CD spectra of *trans*-(*in*-R_p/*out*-S_p)-MUJs is closely related to solvation modes and arrangement of phenyl rings of pillar[n]arene. For **MUJ2**, the host-guest binding effects between the solvent molecules acetonitrile, dichloromethane, tetrahydrofuran, *n*-hexane and the cavity of pillar[5]arene facilitated the phenyl rings of pillar[5]arene array more regularly, while in chloroform, decahydronaphthalene and methanol, except different solvation effects, the phenyl rings of **MUJ2** could also be partially oblique to present a weaker CD signal.

Page 8, lines 117-120 “On the other hand, CD signals of *trans*-(*in*-R_p/*out*-S_p)-**MUJ2** in acetonitrile, dichloromethane, and *n*-hexane were stronger than in chloroform, decahydronaphthalene, and methanol, presumably due to the strong complexation of the formers with pillar[5]arene which fixes the orientation of each subunit.” has been added.

5) In the legend of Figure 7, “(c) Schematic diagram for the temperature-limit-protected light-driven chirality switch (*in*-R_p/*out*-S_p)-**MUJ1**” should be removed.

Action: The sentence “(c) Schematic diagram for the temperature-limit-protected light-driven chirality switch (*in*-R_p/*out*-S_p)-**MUJ1**” has been removed from the description of Fig 7.

Reviewer #2 (Remarks to the Author):

This is an interesting manuscript that describes a molecular switch based on pillar[n]arenes with an appended diazo-benzene containing loop. This structure allows photoinduced chiroptical switching (known property of this design) to be prohibited when the temperature exceeds a

critical limit. The authors describe this as the first realization of overtemperature protection for a molecular photo-switch. The upper-temperature limit was shown to be adjustable by manipulating the solvent. The experiments are detailed and complete, while the rationales/conclusions are appropriate. I recommend publication after attention to the following comments.

We thank this reviewer for his/her highly positive comments.

1. The introductory paragraph describes examples and defines molecular machines in rather broad terms. Some of this is not very precise. For example, machines do work while switches (like those in this article) do not; this is a very important distinction. This paragraph and the associated references should be checked and reworded so as not to perpetuate any incorrect or vague terminology.

We thank the reviewer for the professional and valuable suggestions.

Action: In the introductory paragraph, “molecular machine” were replaced with “molecular machine/device”.

2. I encourage updating of references for certain topics, e.g. Molecular pumps *J. Am. Chem. Soc.* 2020, 142, 34, 14443 and *J. Am. Chem. Soc.* 2019, 141, 44, 17472, Molecular shuttles *Nat. Chem.* 2015, 7, 514 and *Nat. Chem.* 2018, 10, 625.

Action: The two references “Guo, Q.-H. *et al.* Artificial molecular pump operating in response to electricity and light. *J. Am. Chem. Soc.* **142**, 14443-14449 (2020)” and “Qiu, Y. *et al.* A molecular dual pump. *J. Am. Chem. Soc.* **141**, 17472-17476 (2019)” have been cited in the description of molecular “pumps”. And the two references “Zhu, K., O’Keefe, C. A., Vukotic, V. N., Schurko, R. W. & Loeb, S. J. A molecular shuttle that operates inside a metal–organic framework. *Nat. Chem.* **7**, 514-519 (2015)” and “Zhu, K., Baggi, G. & Loeb, S. J. Ring-through-ring molecular shuttling in a saturated [3]rotaxane. *Nat. Chem.* **10**, 625-630 (2018)” have been cited in the description of molecular “shuttles”.

3. When varying temperature for measurements, this should be referred to as variable temperature and abbreviated as VT (e.g. VT NMR).

Action: “Varying-temperature” and “Temperature-dependent” have been replaced by “Variable temperature”, and “V-T” has been replaced as “VT” in the main text.

4. Fig. 7, there is no part (c) as described in the caption.

Action: The sentence “(c) Schematic diagram for the temperature-limit-protected light-driven chirality switch (*in-R_p/out-S_p*)-**MUJ1**” has been removed from the legend of Figure 7.

5. HRMS is high resolution mass spectrometry – not spectroscopy.

Action: We have deleted “spectroscopic” from the sentence “The chemical structure of the MUJs was characterized by HR-mass and NMR spectroscopic studies”.

6. Scheme-1: In (a) the chemical structure would be easier to view without the background graphic (which is not described or particularly relevant), please remove. This will make it easier also to

understand the relationship between the chemical structure and the cartoons used in (b) and the rest of the article.

Action: The background graphic in scheme 1a has been removed.

7. Figure 1: In (a) the m and brackets should be removed from the cis structure. The trans-cis conversion is shown as out-out in (a) but for (b) the cartoons show trans as in? Which is correct? Is the in-out a fast (NMR timescale) equilibrium? This equilibrium (Scheme-1b) should be described in more detail first before the cis trans isomerization. For example, the ¹H NMR spectra shown in (b) are certainly indicative of a reversible cis-trans isomerization but cannot (by themselves) distinguish between trans in or out. I understand that the HPLC resolution and CD spectral assignments address the 4 possible isomers, but it is not clear how these mixtures manifest themselves in the simple ¹H NMR shown in (b).

We thank this reviewer for his/her careful reading and valuable suggestions.

Action:

1. The “m” and brackets have been removed from the *cis* structure in Fig. 1a.
2. It should be *cis-out/trans-in* switch, and we have revised the cartoon’s structures in Fig 1a.
3. The in-out equilibrium of **MUJ1** is a fast equilibrium on the NMR timescale, and we can observe only one set NMR signals of trans-isomers.
4. At the end of the second paragraph, we have added the description of the in-out equilibrium “As shown in scheme 1b, the self-included “*in*” and self-excluded “*out*” conformers of the *trans*-MUJ enantiomers can interconvert quickly accompanying by the switching of chiral conformers that have clockwise-directed (*R_p* conformer) and anticlockwise-directed (*S_p* conformer) arrangement of the substituents, respectively.^{36,37} However, for *cis*-MUJ, the large steric hindrance of *cis*-azobenzene prevents the self-inclusion by the cavity of pillar[6]arene and inhibits the *in-out* interconversion”.
8. It would be appropriate to have the host guest interactions for G1 and G2 be more quantitative. What are the association constants for these interactions? Can the *K_{assoc}* for the in versus out isomers be measure? How do the *K_{assoc}* values in different solvents correlate to the easy of switching.

We thank the reviewer for the helpful suggestions and discussions.

Action:

1. The CD spectra of *trans-(in-R_p/out-S_p)-MUJ2* with **G2** (Fig 3b) and *trans-(in-R_p/out-S_p)-MUJ3* with **G1** (Fig 3c) have been replaced by that having more exquisite titration.
2. The association constants are $(4.9 \pm 0.2) \times 10^3$, $(8.0 \pm 0.8) \times 10^4$ and $(7.5 \pm 0.4) \times 10^3$ for **MUJ1** with **G1** (Supplementary Fig. 42), **MUJ2** with **G2** (Supplementary Fig. 44) and **MUJ3** with **G1** (Supplementary Fig. 46) respectively. Page 6, lines 97-99: “Based on the titration experiments (Supplementary Fig. 42, 44 and 46) the association constants *K_{assoc}* of **G1** were determined as $(4.9$

$\pm 0.2 \times 10^3 \text{ M}^{-1}$ for **MUJ1** and $(7.5 \pm 0.4) \times 10^3 \text{ M}^{-1}$ for **MUJ3**, respectively, and **G2** showed a K_{assoc} of $(8.0 \pm 0.8) \times 10^4 \text{ M}^{-1}$ for **MUJ2**.” has been added.

3. The in/out conformers of MUJs can interconvert very quickly, and we couldn't separate these conformers and were not able to determine the K_{assoc} for the *in* versus *out* isomers.

4. There is no specific host-guest interaction of discrete *trans*-azobenzene with pillar[n]arene due to lack of mainly driving force (Figures 3-4 in appendix). However, when azobenzene is linked to pillar[6]arene, the switching of the side ring of MUJs should be mainly controlled by the equilibrium between the solvation effect and the noncovalent interactions with the cavity of pillar[6]arene. So higher K_{assoc} values of guest molecules with pillar[n]arene in different solvents should facilitate the formation of the penetrated host-guest complex to give *out* conformers. We have added the schematic illustration for the host-guest complexing triggered in-out equilibrium shift of MUJs (Fig 3).

9. Do not abbreviate n-hexane to n-H, or acetonitrile to AN, likewise CHL, DCM etc. Use chemical formulae or accepted abbreviations e.g. MeCN for acetonitrile.

Action:

The solvent abbreviations in the manuscript and the supplementary file have been replaced by full names, i.e. MeOH = methanol, AN = acetonitrile, CHL = chloroform, DCM = dichloromethane, THF = tetrahydrofuran, DECA = decahydronaphthalene, n-H = n-hexane.

10. Describing switching in a thin film as the switching in the solid-state is not accurate. This is an ill-defined, amorphous film not a well packed solid.

Action:

The descriptions “solid-state” have been replaced by “coating film”.

11. Is Fig.7 not misleading? The chemical drawings only show cis as out and trans as in, but these are equilibria and the T dependence is actually the shifting of this equilibrium with change in T. Please define coloured traces.

Action:

We thank the reviewer for the professional and helpful comments.

1. We have added the temperature dependent in-out equilibrium in the schematic of Fig. 7a.
2. We have defined coloured traces in the legend of Fig 7, i.e. “The CD spectra of (*in*- R_p /*out*- S_p)-**MUJ1** with positive CD extrema (CD_{ex}) at ca. 310 nm is shown as the red line, while that with negative CD_{ex} is shown as the pink line and that between the two is shown as the magenta line” for legend (a) and “The blue signet presenting negative g factor and the red signet presenting positive g factor” for legend (b).

12. Perhaps g factor could be defined for those not familiar with this treatment of CD spectra.

Action: Page 9, lines 140-141: “anisotropy factor $g = \Delta\epsilon/\epsilon$, where ϵ is the molar extinction coefficient at a particular wavelength” has been added.

Reviewer #3 (Remarks to the Author):

In this publication by Yang et al., an inherently chiral pillar[n]arene macrocycle (R/S) is strapped by an azobenzene photoswitch (cis/trans), a unit which can either bind inside or outside (in/out) the macrocycle's cavity. As indicated by the brackets, this system contains multiple switching components and their relationship to one another are explored in detail through this work. The key findings of this work are:

1. The isomer of the azobenzene (cis/trans) influences azobenzene binding inside the cavity
2. The inclusion of the azobenzene strap (as trans) can be controlled by light and switches the chirality of the pillar[n]arene (from S to R)
3. MUJ1 is the 'goldlocks' system, with optimal sterics and flexibility for effective switching
4. Temperature is also a stimulus because the light-induced chiral switch only occurs within a strict temperature range, i.e. the system is 'on' at low T and 'off' at high T
5. This 'temperature gating' (i.e. turning the system off at high T) is novel and advantageous because typically a higher T will enhance dynamic processes (turn on). The authors ascribe this to 'over temperature' protection.

In general, I agree with the authors on the novelty of the above. However, there are some key citations (one missing) that overlap and are relevant to the key findings above. Missing citations must be included and all should be better contextualised for this work. I have numbered these in relation to the above:

1. Differences in cis/trans azobenzene binding by pillar[n]arenes has been documented in this publication: J. Am. Chem. Soc. 2012, 134, 20, 8711–8717. Note that this is for a host-guest system and not a '[1]catenene' system described here.
2. Chirality switching of pillar[n]arene has been achieved through guest binding (Chem. Commun., 2020,56, 8424-8427 *missing*), solvent binding (Angew. Chem. Int. Ed. 2013, 52, 8111 –8115) and acid/base (J. Am. Chem. Soc. 2020, 142, 46, 19772–19778). To the best of this reviewers knowledge the work here is the first example of light switchable chirality in a pillar[n]arene.

I am confident that both supramolecular and chiroptic communities will find this work interesting. The ability to switch chirality using a convenient achiral stimuli (light/temperature) is clearly important. Furthermore, as the authors note, the chirality switching results in a switch in CD sign which is more advantageous than intensity changes. This widens the applicability of this work to, for example, sensing and organic materials communities.

Overall, the work performed is convincing and the authors should be congratulated on the thoroughness of this study. However, this reviewer would ask that the following experiments be considered in order to strengthen the conclusions.

We thank this reviewer for his/her highly positive comments.

1. In lines 118-119 authors present the logical conclusion that trans-azobenzene binds in the cavity but cis cannot. Can the authors confirm this through host-guest binding studies, using an

azobenzene as a discrete guest molecule for the cavities of the pillar[n]arenes? This work would link to the previous publication in this area: J. Am. Chem. Soc. 2012, 134, 20, 8711–8717.

In connection with line 183 and following on from the above, the authors should perform variable temperature analysis of the binding of cis/trans-azobenzene guests to the pillar[n]arene hosts to elucidate the thermodynamic contributions to binding. This would provide clear evidence to support their conclusion that an entropic penalty is imposed upon guest binding. This is an important experiment because this entropic barrier is key to the temperature gating of the chiroptical switching behavior.

Action:

We thank the reviewer for the valuable suggestions.

1. We have measured the ^1H NMR spectra of diethyl pillar[6]arene (**DEP6**) and diethyl pillar[5]arene (**DEP5**) in the absence and presence of equivalent *trans*-azobenzene in CDCl_3 (Figures 3-4 in appendix), no chemical shift change of **DEP5** or **DEP6** as well as *trans*-azobenzene could be observed. These results suggested that there is no specific host-guest binding between pillar[n]arene and discrete *trans*-azobenzene. We also have tried the NMR titration of **DEP6** with discrete *trans*-azobenzene in the nonpolar solvent (methyl cyclohexane- d_{14}). However, the poor solubility (< 1 mM) of **DEP6** prevented us from measuring the host-guest interaction of **DEP6** with *trans*-azobenzene in methylcyclohexane- d_{14} . The weak interaction between **DEP6** and *trans*-azobenzene seems reasonable, since different from the cationic azobenzene derivative that used in the previous publication (J. Am. Chem. Soc. 2012, 134, 20, 8711–8717), the azobenzene in our work is neutral, which lacks sufficient driven force to form a stable host-guest complex with **DEP6**. We believe that π - π interaction and the solvophobic interaction should play a role in forming the self-inclusion conformers of MUJs. Therefore the side-ring of **MUJ1** prefers to be accommodated into the cavity of **DEP6** in *n*-hexane, decahydronaphthalene and methanol (Fig 3a).
2. Though there is no specific host-guest binding of discrete *trans*-azobenzene with **DEP6**, we can directly conduct the VT CD experiment of *trans*-(*in*- R_p /*out*- S_p)-**MUJ1** to elucidate the thermodynamic contributions to the in-out equilibrium shift (Supplementary Scheme 2), and $\Delta\Delta H = -4.5$ KJ mol $^{-1}$, $\Delta\Delta S = -16.5$ J mol $^{-1}$ K $^{-1}$ were obtained for the shifting of *out* to *in* chiral inversion in chloroform (Supplementary Fig 107-110). (Chem. Eur. J. 2019, 25, 12526-12537)

Page14, lines 206-209: “Based on the VT CD spectral changes,⁴³ the entropy ($\Delta\Delta S$) and enthalpy ($\Delta\Delta H$) changes for the in/out conformational switching of **MUJ1** in CHCl_3 were estimated as -16.5 J mol $^{-1}$ K $^{-1}$ and -4.5 KJ mol $^{-1}$, respectively, demonstrating that the *out*-to-*in* conformational switching is entropy unfavored by enthalpy preferred.” has been added.

2. The authors conclude that the conformation of the system is solvent dependent (lines 89-90). Therefore it is important that the UV-Vis and NMR data reported in Fig 1 are also performed in the same solvent for accurate comparisons, e.g. UV-Vis to be performed in CHCl_3 .

Similarly, line 93 discusses the potential for ‘binding of the solvent molecules’. Do the authors have evidence for DCM/CHCl₃ encapsulation within the pillar[n]arene instead of the thread? Perhaps the above UV-Vis and NMR spectroscopic experiments suggested above could shed light. Furthermore, reference to the following is essential: *Angew. Chem. Int. Ed.* 2013, 52, 8111–8115 and *Chem. Commun.*, 2020,56, 8424-8427.

Action:

We thank the reviewer for the professional and helpful suggestions.

1. The UV-Vis spectrum in Fig1 has been replaced by the UV-Vis spectrum carried out in chloroform.

2. According to previous reports (*New J. Chem.*, **2014**, 38, 845-851; *Chem. Eur. J.*, **2019**, 25 , 12526-12537), dichloromethane can be encapsulated into the cavity of pillar[5]arene with a binding constant $K_a = 7.9$ in CDCl₃ and 50.7 in methylcyclohexane-d₁₄. However, the host-guest interaction of CHCl₃ with pillar[5]arene is too small to measure due to the mismatch of molecular size (*J. Phys. Chem. B.* **2015**, 119, 6711–6720; *Chem. Sci.*, **2021**, DOI: 10.1039/D0SC06988D).

The poor solubility (< 1 mM) of diethyl pillar[6]arene in nonpolar solvent (methyl cyclohexane-d₁₄) makes it difficult to measure the binding constant of pillar[6]arene with CHCl₃ by NMR titration. We tried UV-Vis spectra titrations of pillar[6]arene with CHCl₃ and CH₂Cl₂ in methyl cyclohexane, but observed little change on the spectra (Appendix, Figures 9-10), suggesting very weak, if any, host-guest interaction of CHCl₃ and CH₂Cl₂ with pillar[6]arene. This seems reasonable considering the fact that the cavity of pillar[6]arene is too large relative to the size of CHCl₃ and CH₂Cl₂ molecules.

3. The references ref. 36 (“Nagata, Y. *et al.* Holding of planar chirality of pillar[5]arene by kinetic trapping using host–guest interactions with achiral guest solvents. *Chem. Commun.* **56**, 8424-8427 (2020)”) and ref. 37 (“Ogoshi, T., Akutsu, T., Yamafuji, D., Aoki, T. & Yamagishi, T.-a. Solvent- and achiral-guest-triggered chiral inversion in a planar chiral pseudo[1]catenane. *Angew. Chem. Int. Ed.* **52**, 8111-8115 (2013)”) have been cited in the description of the chirality of *trans*-MUJ in the second paragraph.

3. Is it possible to conduct the HPLC shown in Fig 2 a with the cis-isomerised azobenzene to shown an inversion of the peak intensities i.e. b,c >>> a,d.

Action:

1. Yes, the HPLC peaks show an intensity inversion of the peaks a/d and b/c (Figure 11 in appendix), i.e. peaks a/d decreased and b/c increased after 365 nm photoirradiation.

Page 5, lines 67-68: “Photoirradiation of the mixture at 365 nm led to an inversion of the relative intensities of the peaks a/d and b/c (Supplementary Fig. 41).” has been added.

Some further points that are more minor but would greatly improve the publications clarity for the reader are listed below:

- Line 9: I think ‘large-amplitude’ should be added when deifying the molecular motion in molecular machines

Action: “large-amplitude” has been added to the front of “mechanical motions” in line 9.

- Scheme 1: Please indicate the planar chirality that arises from a pillar[n]arene on the cartoon (e.g. an arrow?). It might not be instantly obvious where the source of chirality is in the system and to separate it from the cis/trans geometry of the azobenzene.

Action: Following the reviewer's good suggestion, we have added the arrow to indicate the planar chirality of pillar[n]arene in scheme 1b. At the end of the second paragraph, we have added the description of the in-out equilibrium “As shown in scheme 1b, the self-included “*in*” and self-excluded “*out*” conformers of the *trans*-MUJ enantiomers can interconvert quickly accompanying by the switching of chiral conformers that have clockwise-directed (R_p conformer) and anticlockwise-directed (S_p conformer) arrangement of the substituents, respectively. However, for *cis*-MUJ, the large steric hindrance of *cis*-azobenzene prevents the self-inclusion by the cavity of pillar[6]arene and inhibits the *in-out* interconversion”.

- Lines 36-38: The ¹H NMR shifts are indicating more than just cis-trans isomerization. It is important to clarify that it indicated major changes in the molecule's conformation i.e. in/out

Action: Page 3 lines 42-46: “Meanwhile, proton *b* of the side ring-bearing hydroquinone unit showed a downfield shift (Fig. 1b) while upfield shift could be observed with protons of several other hydroquinone units, suggesting a significant conformational change from self-included *in* to self-excluded *out* conformation upon the *trans* to *cis* isomerization which released the shielding and deshielding effects unequivocally exerted on pillar[6]arene subunits” has been added.

- Line 40: Clarify to the reader that the lifetime of *cis* isomer is discussed later

Figure 1: There should be consistency in representation of the major conformation of the macrocycle in *cis/trans* isomer i.e. in *trans* the azobenzene is included in the macrocycle and should be indicated in Fig 1a structure (as it is in Fig 1b)

Action:

1. The sentence “The lifetime of *cis* isomers of MUJs will be discussed later” has been added to the last of the third paragraph of the main text.

2. We have modified the structures in Fig 1a accordingly.

- Line 53: The planar chirality arising from the pillar[n]arene is highlighted but should be indicated to the reader earlier in the manuscript (e.g. lines 26-28)

Action:

We thank the reviewer for the helpful suggestion.

We have added the description of the in-out equilibrium as well as the chirality of pillar[n]arene to the end of the second paragraph, i.e. “As shown in scheme 1b, the self-included “*in*” and self-excluded “*out*” conformers of the *trans*-MUJ enantiomers can interconvert quickly accompanying by the switching of chiral conformers that have clockwise-directed (R_p conformer)

and anticlockwise-directed (S_p conformer) arrangement of the substituents, respectively. However, for *cis*-MUJ, the large steric hindrance of *cis*-azobenzene prevents the self-inclusion by the cavity of pillar[6]arene and inhibits the *in-out* interconversion”.

- Line 71: It would be helpful for the reader to see a cartoon of how guest binding leads to the strap being ejected i.e. favouring an out conformation

Action:

We thank the reviewer for the helpful suggestion.

We have added the cartoon graphic in Fig.3 to illustrate the host-guest complexing triggered in-out equilibrium shift of MUJs.

- Line 90: For clarity please adjust to read ‘conformation dominated in all these solvents’

Action: “conformation dominated in these solvents” in the main text has been adjusted to “conformation dominated in all these solvents”.

- Fig 4 would benefit from a cartoon to show the reader what conformational and chiral changes these CD spectra are evidencing

Action: We have added cartoons to the CD spectra of Fig 4 to present the conformational and chiral changes.

- Line 119: Supplementary Scheme 2 should be brought into the manuscript at this point to add clarity for the reader.

Action:

The schematic diagram in Supplementary Scheme 2 has been brought into the manuscript showing as the Scheme 2.

- Line 180 (and all solvent abbreviations): Please can solvents be written in full e.g. CHL = CHCl₃ or chloroform

Action:

We thank the reviewer for the helpful suggestion.

The solvent abbreviations in the manuscript and supplementary file have been written in full name, i.e. MeOH = methanol, AN = acetonitrile, CHL = chloroform, DCM = dichloromethane, THF = tetrahydrofuran, DECA = decahydronaphthalene, *n*-H = *n*-hexane.

The experimental details provided in the supporting information are detailed and thorough, suggesting that this work could be reproduced.

There is considerable interest in the development of chiroptical switches that can be controlled using facile external stimuli such as light. This publication may influence thinking in this area by asking researchers to consider the influence of entropy and thus temperature as a stimulus that is competitive or, perhaps in the future, can be harnessed.

Thanks again to the reviewer for his very professional and meaningful suggestions and comments.

Appendix

Figure 1. Left: CD spectra of *trans*-(*in*-R_p/*out*-S_p)-MUJ2 (0.038 mM) in the presence of various amounts of G2 (0~4.8 equivalents) in CHCl₃ at 25 °C. Right: Association constant of MUJ2 with G2 fitted based on CD titration.

Figure 2. Left: CD spectra of *trans*-(*in*-R_p/*out*-S_p)-MUJ3 (0.044 mM) in the presence of various amounts of G1 (0~12.1 equivalents) in CHCl₃ at 25 °C. Right: Association constant of MUJ3 with G1 fitted based on CD titration.

Scheme 1. Schematic illustration for the host-guest complexing triggered in-out equilibrium shift of MUJs.

Figure 3. ^1H NMR spectra of (a) **DEP5** (22 mM, 400 MHz, rt) and (b) the 1:1 mixed solution of **DEP5** (22 mM) with *trans*-azobenzene (22 mM) (b) and (c) *trans*-azobenzene (22 mM) in CDCl_3 .

Figure 4. ^1H NMR spectra of (a) **DEP6** (22 mM, 400 MHz, rt) and (b) the 1:1 mixed solution of **DEP6** (22 mM) with *trans*-azobenzene (22 mM) (b) and (c) *trans*-azobenzene (22 mM) in CDCl_3 .

Thermodynamic parameters of the out-to-in equilibrium

Scheme 2. Schematic representations of (a) the in-out equilibrium of *trans*-(*in*- R_p /*out*- S_p)-**MUJ1** and (b) the penetrative host-guest complex of *trans*-(*in*- R_p /*out*- S_p)-**MUJ1** with **G1**.

$$(1) y_{\text{in}}(-g_{100\% \text{out}}) + (1-y_{\text{in}})g_{100\% \text{out}} = g$$

$$y_{\text{in}} = (1-g/g_{100\% \text{out}})/2$$

$$(2) K_{in/out} = y_{in}/(1-y_{in})$$

Figure 5. VT CD spectra of (a) *trans*-(*in*-*R_p*/*out*-*S_p*)-**MUJ1** and (b) *trans*-(*in*-*R_p*/*out*-*S_p*)-**MUJ1** with excessive **G1** in chloroform.

Figure 6. VT UV-Vis spectra of (a) *trans*-(*in-R_p*/*out-S_p*)-MUJ1 and (b) *trans*-(*in-R_p*/*out-S_p*)-MUJ1 with excessive G1 in chloroform.

Figure 7. VT dissymmetry factor (g) spectra of (a) *trans*-(*in*- R_p /*out*- S_p)-**MUJI** and (b) *trans*-(*in*- R_p /*out*- S_p)-**MUJI** with excessive **G1** in chloroform.

Figure 8. van't Hoff plot for the in-out equilibrium of *trans*-(*in*- R_p /*out*- S_p)-**MUJI** at different temperature in chloroform, and thermodynamic parameters $\Delta\Delta H = -4.5 \text{ KJ mol}^{-1}$, $\Delta\Delta S = -16.5 \text{ J mol}^{-1} \text{ K}^{-1}$ are obtained for the *out* \square to *in* chiral inversion.

Figure 9. UV-Vis spectra of **DEP6** with titrations of CHCl_3 in methyl cyclohexane.

Figure 10. UV-Vis spectra of **DEP6** with titrations of CH_2Cl_2 in methyl cyclohexane.

Figure 11. Chiral-phase HPLC traces of **MUJ1** (*n*-hexane : tetrahydrofuran = 4 : 1) before and after 365 nm photoirradiation (xenon grating spectrometer, 10 min), detecting at 295 nm (black) and 319 nm (red), respectively.

REVIEWERS' COMMENTS

Reviewer #1 (Remarks to the Author):

This revised manuscript can be published as it is now.

Reviewer #2 (Remarks to the Author):

This is a much better manuscript and is (almost) ready for publication. Thanks you for your extra efforts to make it so. Only one item requires further attention.

5. HRMS is high resolution mass spectrometry – not spectroscopy.

Action: We have deleted “spectroscopic” from the sentence “The chemical structure of the MUJs was characterized by HR-mass and NMR spectroscopic studies”.

Reviewers Comments: Unfortunately, this amendment still infers that HR-mass is a spectroscopy. Please use "The chemical structure of the MUJs was characterized by HR-mass spectrometry and NMR spectroscopic studies.

Reviewer #3 (Remarks to the Author):

This reviewer believes the authors have done an excellent job in addressing all the points that have been raised. In particular it is key that the switching process was demonstrated to be disfavoured entropically. As a final request, please could the authors tighten up their language at this point: line 209 and 210 on page 14.

"demonstrating that the out-to-in conformational switching is entropy unfavored by enthalpy preferred".

Please consider changing this to a new sentence that reads: "This demonstrates that the out-to-in conformational switching is enthalpically favourable but disfavoured entropically."

Reference to the importance of this for the temperature gating might also be included.

Once addressed then I would recommend this work for publication in Nature Communications.

Response to the reviewer comments

Reviewer #1 (Remarks to the Author):

This revised manuscript can be published as it is now.

Action: We thank this reviewer for his/her highly positive comments.

Reviewer #2 (Remarks to the Author):

This is a much better manuscript and is (almost) ready for publication. Thanks you for your extra efforts to make it so. Only one item requires further attention.

5. HRMS is high resolution mass spectrometry – not spectroscopy.

Action: We have deleted “spectroscopic” from the sentence “The chemical structure of the MUJs was characterized by HR-mass and NMR spectroscopic studies”.

Reviewers Comments: Unfortunately, this amendment still infers that HR-mass is a spectroscopy. Please use "The chemical structure of the MUJs was characterized by HR-mass spectrometry and NMR spectroscopic studies.

Action:

We thank this reviewer for his/her highly positive comments and helpful suggestion.

Page 2 lines 41-42, The sentence “The chemical structure of the MUJs was characterized by HR-mass and NMR studies” has been replaced by “The chemical structure of the MUJs was characterized by HR-mass spectrometry and NMR spectroscopic studies”.

Reviewer #3 (Remarks to the Author):

This reviewer believes the authors have done an excellent job in addressing all the points that have been raised. In particular it is key that the switching process was demonstrated to be disfavoured entropically. As a final request, please could the authors tighten up their language at this point: line 209 and 210 on page 14.

"demonstrating that the out-to-in conformational switching is entropy unfavored by enthalpy preferred".

Please consider changing this to a new sentence that reads: "This demonstrates that the out-to-in conformational switching is enthalpically favourable but disfavoured entropically."

Reference to the importance of this for the temperature gating might also be included.

Once addressed then I would recommend this work for publication in Nature Communications.

We highly appreciate this reviewer for his very professional and meaningful suggestions and comments.

Action: Page 15 lines 223-224, The sentence “demonstrating that the out-to-in conformational

switching is entropy unfavored by enthalpy preferred” has been replaced by “This demonstrates that the out-to-in conformational switching is enthalpically favourable but disfavoured entropically”.